# Cannabidiol and Vitamin D3 Impact on Osteogenic Differentiation of Human Dental Mesenchymal Stem Cells

**DOI:** 10.3390/medicina56110607

**Published:** 2020-11-12

**Authors:** Nausica B. Petrescu, Ancuta Jurj, Olga Sorițău, Ondine P. Lucaciu, Noemi Dirzu, Lajos Raduly, Ioana Berindan-Neagoe, Mihai Cenariu, Bianca A. Boșca, Radu S. Campian, Aranka Ilea

**Affiliations:** 1Department of Oral Health, University of Medicine and Pharmacy “Iuliu Hatieganu”, Victor Babes Street, No. 15, 400012 Cluj-Napoca, Romania; nausica_petrescu@yahoo.com (N.B.P.); rcampian@email.com (R.S.C.); 2Research Center for Functional Genomics, Biomedicine and Translational Medicine, “Iuliu Hatieganu” University of Medicine and Pharmacy, Gheorghe Marinescu Street, No. 23, 400337 Cluj-Napoca, Romania; ancajurj15@gmail.com (A.J.); raduly.lajos78@gmail.com (L.R.); ioananeagoe29@gmail.com (I.B.-N.); 3Department of Functional Genomics and Experimental Pathology, “Prof. Dr. Ion Chiricuta” Oncology Institute, Republicii Street, No. 34-36, 400015 Cluj-Napoca, Romania; 4Radiotherapy, Radio-biology and Tumor Biology Laboratory, The Oncology Institute “Prof. dr. Ion Chiricuta”, Republicii Street, No. 34-36, 400015 Cluj-Napoca, Romania; olgasoritau@yahoo.com; 5Research Center for Advanced Medicine, MedFuture, University of Medicine and Pharmacy “Iuliu Hatieganu”, Louis Pasteur Street, No, 4, 400000 Cluj-Napoca, Romania; noemidirzu@gmail.com; 6Department of Clinical Sciences, Faculty of Veterinary Medicine, University of Agricultural Sciences and Veterinary Medicine, Calea Manastur, No. 3-5, 400372 Cluj-Napoca, Romania; mihai.cenariu@usamvcluj.ro; 7Department of Histology, Faculty of Medicine, “Iuliu Haţieganu” University of Medicine and Pharmacy, Louis Pasteur Street, No. 6, 400349 Cluj-Napoca, Romania; biancabosca@yahoo.com; 8Department of Oral Rehabilitation, University of Medicine and Pharmacy “Iuliu Hatieganu”, Victor Babes street, No. 15, 400012 Cluj-Napoca, Romania; arankailea@yahoo.com

**Keywords:** dental stem cells, cannabidiol, Vitamin D, osteogenesis, bone regeneration, tissue engineering, CB2

## Abstract

*Background and objective:* The aim of the present study was to establish a new differentiation protocol using cannabidiol (CBD) and vitamin D3 (Vit. D3) for a better and faster osteogenic differentiation of dental tissue derived mesenchymal stem cells (MSCs). *Materials and methods:* MSCs were harvested from dental follicle (DFSCs), dental pulp (DPSCs), and apical papilla (APSCs) of an impacted third molar of a 17-year old patient. The stem cells were isolated and characterized using flow cytometry; reverse transcription polymerase chain reaction (RT-PCR); and osteogenic, chondrogenic, and adipogenic differentiation. The effects of CBD and Vit. D3 on osteogenic differentiation of dental-derived stem cell were evaluated in terms of viability/metabolic activity by alamar test, expression of collagen1A, osteopontin (OP), osteocalcin (OC), and osteonectin genes and by quantification of calcium deposits by alizarin red assay. *Results:* Stem cell characterization revealed more typical stemness characteristics for DFSCs and DPSCs and atypical morphology and markers expression for APSCs, a phenotype that was confirmed by differences in multipotential ability. The RT-PCR quantification of bone matrix proteins expression revealed a different behavior for each cell type, APSCs having the best response for CBD. DPSCs showed the best osteogenic potential when treated with Vit. D3. Cultivation of DFSC in standard stem cell conditions induced the highest expression of osteogenic genes, suggesting the spontaneous differentiation capacity of these cells. Regarding mineralization, alizarin red assay indicated that DFSCs and APSCs were the most responsive to low doses of CBD and Vit. D3. DPSCs had the lowest mineralization levels, with a slightly better response to Vit. D3. *Conclusions:* This study provides evidence that DFSCs, DPSCs, and APSCs respond differently to osteoinduction stimuli and that CBD and Vit. D3 can enhance osteogenic differentiation of these types of cells under certain conditions and doses.

## 1. Introduction

Bone defects are one of the main challenges in dentistry and in maxillofacial and orthopaedic surgery. Bone defects can have infectious, traumatic, or congenital origin and they have serious consequences on the patients’ quality of life, with a negative social and economic impact. In dentistry, bone defects are caused by dental extractions, periodontal disease, peri-implantitis, apicoectomy, cystectomy, atrophy of the maxillary or mandibular alveolar bone, resection of the maxillary or mandibular bone, osteomyelitis [1]. Even though the bone has an impressive regeneration potential, in severe cases, spontaneous healing is impossible and the treatment methods for managing bone defects are difficult to acquire and not always successful. This context enquires for new procedures to stimulate bone regeneration [2]. Stem cells are a very suitable tool for producing and stimulating bone regeneration [3]. In the last decade, the implication of stem cells in tissue regeneration was proven and investigated in vitro and in vivo [4,5]. The multipotent capacity of adult mesenchymal stem cells (MSCs) can be proven by their ability to differentiate into multiple other cell lines: osteoblasts, adipocytes, chondrocytes [6] in vitro by using specific culture media with added growth factors. MSCs can be isolated from different types of tissues from the oral cavity, such as dental pulp, apical papilla, dental follicle, periodontal ligaments, and exfoliated deciduous teeth [6]. Of great interest, dental tissues possess an advantage in terms of ease harvesting and lack of ethical implications for use in tissue engineering. MSCs isolated from dental tissues exhibit self-renewal, multilineage differentiation potential, and immunomodulatory properties, hence the possibility of using them in regenerative medicine and immunotherapies, as several studies have demonstrated [7,8]. Generally, isolated cells from these tissues show the basic characteristics of mesenchymal adult stem cells. They have a fibroblast like morphology and express several specific surface markers: CD105, CD73, CD90, with a variable expression level for CD44, CD71, CD271, CD166, CD29 [1]. Even though dental-tissue derived MSCs share common features, they can express different phenotypic, genotypic, and protein expression patterns, which can then be reflected in distinct functional properties, such as in their differentiation potential [9]. However, they can also have some limitations because there are more specialized tissues that do not exhibit a continuous remodeling as the bony tissue. Consequently, stem/progenitor cells from those tissues may be more restricted in their multipotency in comparison with bone marrow derived mesenchymal stem cells (BM-MSC). However, as an advantage, all dental stem cells (DSCs), particularly APSCs and DFSCs, possessed greater proliferative potential than BM-MSCs as Yuichi Tamaki et al. reported [10,11]. Dental tissues (dental pulp and apical papilla) in the pre-natal development are in early interaction with the neural crest, so stem cells derived from these tissues can share common characters with cells derived from the neural crest, such as a spontaneous expression of neuronal markers even when DPSCs and APSCs were cultured in a medium without neural inductive supplements [12,13]. Collectively, the multipotency, high proliferation rates, and accessibility make the dental stem cells an attractive source of mesenchymal stem cells for tissue regeneration. Bone healing is a topic of great relevance especially in the aging population; the elderly face osteoporosis or osteoarthritis diseases, with a rise in the basal state of inflammation, potentially impairing osteogenesis. Bone regeneration can be described in three phases: inflammatory, repair, and remodeling, phases involving the intervention of several players represented by cells (monocyte-macrophage-osteoclast lineage and the mesenchymal stem cell-osteoblast lineage) and cytokines. This complex interactions between cells and microenvironments (the early inflammatory phase) will determine the activation of several pathways leading to enhanced angiogenesis, proliferation, and differentiation of MSCs towards osteoblasts. [14] An initial inflammatory microenvironment has critical importance because it triggers the osteogenic process cells and induces local immune modulation [14]. From this point of view, the use of therapies that influence both bone regeneration and immunomodulation deserve attention. Vitamin D and Cannabidiol (CBD) are known for their ability to stimulate bone regeneration. For more than 4000 years, cannabis has been suspected to have positive treatment outcomes [15]. In the last few years, its medical potential has been tested on numerous conditions like multiple sclerosis, epilepsy, Parkinson’s Disease, Alzheimer, Huntington’s Disease; it has also been tested as an analgesic, antitumour drug, for brain cancer, breast cancer, prostate cancer, other carcinomas, and other diseases [16]. In the beginning of the 20th century, cannabinoid receptors were confirmed to exist in rat cells: CB1 in 1990 and CB2 in 1993. Both CB1 and CB2 were detected in human osteoblasts and osteoclasts. CB1 receptors are found in both the central and peripheral nervous systems, with the majority of receptors localized to the hippocampus and amygdala of the brain [17]. In the osteoclast cells were also found two other receptors for other endocannabinoids: andamide (arachidonoyl ethanolamide) and 2-arachidonoylglycerol [18]. Over the last few years, many studies have confirmed that CB2 takes part in bone metabolism. It has been demonstrated that CB2 helps regulate bone mass by stimulating osteoblasts and inhibiting osteoclasts [19]. Cannabinoid receptors CB1 and CB2 are known to have an important part in regulating bone metabolism [20,21,22]. CB2 receptors present in osteoblasts, osteocytes, and osteoclasts [19] have a more intense expression than CB1 [22,23] and have also been mentioned to be expressed on bone marrow stem cells [20]. CB1 and CB2 are part of the endocannabinoid system along with their endogenous ligands and enzymes [24]. Tetrahydrocannabinol (THC) and CBD are two of the most used cannabinoid derivates in research. CBD, a major phytocannabinoid, is one of most active cannabinoids identified, accounting for up to 40% of the Cannabis plant’s extract, with molecular formula C21H30O2, IUPAC name 2-[(1R,6R)-3-methyl-6-prop-1-en-2-ylcyclohex-2-en-1-yl]-5-pentylbenzene-1,3-diol [25]. In contrast to ∆9-THC, CBD is a non-psychotropic cannabinoid. CBD stimulates endoplasmic reticulum (ER) stress and inhibits AKT/mTOR signaling, thereby activating autophagy and promoting apoptosis. In addition, CBD enhances the generation of reactive oxygen species (ROS), upregulates the expression of intercellular adhesion molecule 1 (ICAM-1) and tissue inhibitor of matrix metalloproteinases-1 (TIMP1), and decreases the expression of the inhibitor of DNA binding 1 (ID-1) [26]. Although it is known that CBD inhibits bone resorption, the way in which cannabinoids participate in bone reconstruction and stem cell migration have not yet been elucidated [27]. Still, there are studies mentioning, as a future perspective, that targeting CB receptors could be a therapeutic approach for osteoporosis [20]. In recent studies, CBD, gradually released from a scaffold, used for treating bone defects showed positive results [27].

Vitamin D is known to enhance stem cell differentiation to osteoblasts and promote osteogenesis [28,29]. Several studies have speculated on the role of Vit D3 in the differentiation of osteoblasts and, more recently, on mesenchymal stem cells (MSCs), which are known for their abilities in promoting bone repair and regeneration in cell reconstructive therapies. Vitamin D by means of its biological active form, 1α,25-dihydroxyvitamin D3 (1,25(OH)2D3), has a protective effect on the skeleton by acting on calcium homeostasis and bone formation [30]. Therefore, in our study, it was used as a standard osteogenesis promoter in order to measure the effects of CBD.

We propose a new differentiation method of using CBD and Vit. D3 for a better and faster osteogenic differentiation of dental tissue-derived stem cells. MSCs isolated from dental follicle, dental pulp, and apical papilla express mesenchymal stem cells’ markers, with differences between cell types, with a characteristic MSCs phenotype for dental pulp stem cells (DPSCs), an intermediary phenotype for cells derived from dental follicle (DFSCs) and a great cell heterogeneity for apical papilla derived stem cells (APSCs). Treatments of stem cells with CBD, the pharmacologically active phytocannabinoids of Cannabis sativa and with α,25-dihydroxyvitamin D3 (1,25(OH)2D3) (Vit. D3) in long cultivation in standard conditions (8 days) or in osteoinductive medium (12 days) showed no cytotoxicity even at higher doses of 2 mM of CBD. The cell’s response to osteogenic stimuli in combination with CBD or Vit. D3 was different between cell types in terms of expression of bone specific genes (osteopontin, collagen 1A, osteonectin, and osteocalcin) and mineralization. DFSCs showed a spontaneous bone differentiation in standard stem cell medium and simple osteogenic medium. DPSCs had the best osteogenic response to Vit. D3 (2.5 nM) as gene expression but with the lowest mineralization rate (regardless of cultivation conditions). CBD in low dose (0.75 µM) induced osteogenic gene expression and Vit. D3 determined an increased mineralization rate of APSCs.

The purpose of this study is to find a manner of improving the bone regeneration process.

## 2. Materials and Methods

### 2.1. Collection of Tissue Samples

In this study, the mesenchymal dental stem cells derived from dental pulp, apical papilla, and dental follicle were harvested from the same donors’ impacted third molars. The patient had no comorbidities or inflammatory/tumor pathologies involving the oral cavity.

After extraction, the cells were isolated, characterized, and differentiated into osteoblasts under CBD and Vit. D3 exposure.

A 17-year-old female patient was referred to our clinic with four third impacted molars. All third molars had extraction indication due to the lack of space on the dental arches for the teeth to erupt and the dento-maxillary apparatus to function properly. Both the diagnosis and risks of the surgical procedure were explained to the patient and her parents. Informed consent for the teeth extractions and for using the teeth in research purposes was obtained from the patient’s parents, because she was under the age of 18. The study was approved by the Ethics Committee from the “Iuliu Hatieganu” University of Medicine and Pharmacy, Cluj-Napoca, No. 314/12.08.14.

The teeth extractions were performed by an oral surgeon. The teeth were harvested entirely, together with the apical papilla and dental follicle. After the extractions, the teeth were immersed into Dulbecco’s Modified Eagle’s Medium (DMEM) complete cell culture medium, preheated, and maintained at 37 °C and transported in the shortest amount of time to the laboratory.

### 2.2. Stem Cell Isolation

In the cell culture laboratory, the teeth were immersed 15 s in 90% alcohol for disinfection and washed three times with phosphate buffered saline solution (PBS). The following tissues were harvested: dental pulp (DPSCs), apical papilla (APSCs), and dental follicle (DFSCs). The apical papilla was removed with a Gracey curette and the dental pulp was removed from the root canal with a tire-nerf needle. None of the third molars had erupted in the oral cavity, so the dental follicle that surrounds each tooth was also harvested. All three types of tissue were cut into 3–4 mm pieces.

The cells were isolated using the explant technique described by Centeno [31,32]. The tissue fragments were washed three times with PBS. Mechanical dissociation was performed using scissors and scalpel under sterile conditions in a Petri dish, with the tissue fragments immersed in complete medium. In this step, the released cells in medium after mechanical processing were collected with a syringe and the cell suspension was then filtered through Filcons devices (with a 70 μm mesh) to obtain a monocellular suspension and finally centrifuged. The remaining fragments (explants) were also collected in Falcon tube and centrifuged at 1100 rpm, 5 min at room temperature. The fragments and the cell suspension were seeded on 25 cm^2^ Cole flasks in a standard stem cell culture medium consisting of DMEM medium with 4.5 g/L glucose/F12 HAM medium (1:1 ratio), 20% FBS, 1% penicillin + streptomycin, 2 mM L-glutamine, 1% nonessential amino acids (NEA), 1mM sodium pyruvate, and 55 μM β-mercaptoethanol (all reagents were achieved from Sigma Aldrich, St. Louis, MO, USA). In the first 1–2 days, the cells and explants were covered only by a small amount of culture medium to allow the attachment of cells and explants to the plastic surface. Cells were cultured for 10 days in 5% CO2 at 37 °C. Adherent cells that migrated from explants were observed after 4–10 days. The medium was replaced every 2–3 days until the cells had grown to the 70–80% confluency. Trypsinization and cell passage was performed. The cell proliferation rate was different for each type of cell (in relation to the tissue origin), requiring cell passage every 3–6 days. For further experiments, the cells were used at passage 6.

### 2.3. Characterization of the Isolated Cells

#### 2.3.1. Immunocytochemical Characterization by Flow Cytometry

The stemness phenotypic profile of the isolated cells was evaluated at 6th passage for each cell line. When the cells reached 80–90% confluence, cells were detached using 0.25% Trypsin/EDTA (Sigma Aldrich), centrifugated, and counted using a hemocytometer. Cell number was adjusted to 10^5^ cells/sample. Cells were washed twice with PBS and resuspended in 100 μL staining buffer containing 10 μL of each antibody, including positive/negative and single-stain controls. The primary conjugated antibodies dilution was 1:20 for BD antibodies (CD29 PE (Phycoerythrin), CD34 FITC (Fluorescein Isothiocyanate), CD45 FITC, CD49e PE) (BD Biosciences, San Jose, CA, USA), and 1:50 dilution for Santa Cruz Biotechnologies antibodies (Dallas, Texas, U.S.A.) (CD73 PE, CD90 FITC, CD105 PE, CD117 PE) and HLA-DR FITC (Biolegend). Cells were incubated for 45 min at 4 °C in the dark. After incubation, cells were washed twice with ice cold PBS and then analyzed by flow cytometry (BD FACS Canto II flow cytometer; Becton Dickinson, USA) using Diva software.

#### 2.3.2. Reverse Transcription Polymerase Chain Reaction (RT-PCR)

Genetic analyze was performed at 6th passage of cultivation of isolated cells. Total RNA was extracted with TriReagent (Invitrogen) as reported by the manufacturer’s protocol. The RNA was evaluated quantitatively and qualitatively with NanoDrop-1100, by measuring the absorbance of UV-visible light. Total RNA (500 ng) was reversely transcribed into cDNA using the High Capacity cDNA Reverse Transcription Kit (Applied Biosystems) for gene expression evaluation. SYBR Select Master Mix was used for the evaluation of gene expression, and all amplifications and detections were done in the Applied Biosystems ViiA^TM^ 7 System. The primer sequences were as follows in Table 1.

#### 2.3.3. Multipotential Ability-Differentiation Experiments

##### Osteogenic Differentiation

For the osteogenic differentiation, stem cells were seeded on 12 well culture plates at a density of 5 × 10^4^ cells/well in 200 μL/well medium. Simple osteoinductive culture medium (OS medium) was composed of DMEM high glucose/F12 HAM (ratio 1:1) supplemented with 10% FBS (Fetal Bovine Serum), 100 U/mL penicillin, 100 µg/mL streptomycin, 1% NEA, 2 mM L-glutamine, 100 nM dexamethasone, 10 mM β-glycerol phosphate, 0.2 mM L-ascorbic acid 2-phosphate (all Sigma reagents, Sigma-Aldrich). Complex osteoinductive medium (OC medium) was composed by all constituents of OS medium to which was added 10 μg/mL Bone Morphogenic Protein 2 (BMP2). The BMP2 concentration was progressively decreased with each change of medium. The medium was replaced every three days. Twenty-eight days later, the cells were fixed with 4% paraformaldehide for 20 min and stained with alizarin red in order to detect the calcium deposits. After that, they were washed with PBS and exposed to a liquid mixture of 2% alizarin red and deionized water. The next step was vigorous washing of the cells with deionized water and then with PBS. Finally, macroscopic and microscopic images were taken. For quantification, alizarin red deposits were solubilized with 10% cetylpyridinium chloride solution and 200 µL aliquots of purple solution were transferred to a 96 well plate and absorbance was measured with a BioTek Synergy 2 fluorescence microplate reader (Winooski, VT, USA) at 562 nm. The experiments were done in triplicate and statistical analysis was performed, using “Two-way RM ANOVA” with Bonferroni posttest.

##### Immunocytochemical Staining for Osteogenic Markers

The cells were seeded at a density of 4 × 10^4^ cells in 500 μL simple osteogenic medium/well in Nunc Lab-Tek 4 wells chamber slides. After 10 days of cultivation, the cells were washed 3 times with Phosphate Buffered Saline (PBS) and fixed with 4% paraformaldehyde solution for 20 min at room temperature. The cell permeabilization was performed with 0.1% Triton X-100 for 20 min, followed by blocking of unspecific antibody bound with 10% Bovine Serum Albumin (BSA) for 20 min at room temperature. Staining with primary antibodies was performed for bone cells markers: collagen 1A, osteopontin, and osteocalcin (all from Santa Cruz Biotechnologies) at a dilution of 1:50. Samples were incubated overnight at 4 °C in dark, followed by incubation 45 min at room temperature with secondary antibody goat anti-mouse IgG labeled with FITC (Santa Cruz Biotechnologies) (1:50). Slides were mounted with 4,6-diamidino-2-phenylindole (DAPI) (Santa Cruz Biotechnologies) for nuclei staining. Each step was followed by 3 washes with PBS. The samples were visualized with a Zeiss Axiovert fluorescence microscope using filters at 488, 546 and 340/360 and images were taken with an AxioCam MRC camera.

##### Chondrogenic Differentiation

The differentiation medium used contained TGFβ3 growth factor, which is the basic element for chondroblastic differentiation. The differentiation medium was high glucose DMEM/F12 (1:1 ratio) supplemented with 10 μM dexamethasone, 50 μg/mL ascorbic acid, 1 mM sodium pyruvate, 10 ng/mL TGFβ3, 1% ITS (insulin, transferrin, selenium). The cells were seeded on low-attach plates that did not permit cell adhesion and the cells were kept in suspension. After 3 weeks of cultivation, cultures were fixed with 4% paraformaldehide and stained with 3% Alcian Blue solution (Sigma-Aldrich, St. Louis, MO, USA) (pH = 4.1) to identify proteoglycans. Microscopic phase contrast images were taken with a CCD camera (Axiocam MRC) adapted to a Zeiss Axio Observer D1 inverted microscope.

##### Adipogenic Differentiation

MSC cultures were seeded in growth medium on fibronectin-coated (1.5 µg/cm^2^) 12 well plates and grown to confluence. The standard stem cells medium was replaced with differentiation media consisting of DMEM/F12 HAM (Sigma-Aldrich) supplemented with 2 mM L-glutamine, 1% antibiotic, 10-6 M dexamethasone (Sigma-Aldrich, St. Louis, MO, USA), 0.5 mM isobutylmethylxanthine (Sigma-Aldrich), 1% ITS (100× Insulin-Transferrin-Selenium stock from Invitrogen Gibco Life Technologies, Paisley, UK), 200 µM indomethacin, 0.2 nM triiodothyronine (Sigma-Aldrich). The induction medium was replaced every 3 days. After 14 days of culture in adipogenic induction medium, cells were fixed with 4% paraformaldehide for 20 min, washed with PBS, and stained with Oil Red solution (Sigma-Aldrich) for 10 min. Stained cells were visualized under light microscopy using a Axiovert Zeiss microscope. Images were captured with a AxioCam MRC camera.

### 2.4. Evaluation of Canabidiol Effects on Dental Tissues Derived Stem Cells

The study of CBD treatment on isolated stem cells assessed especially the cell response regarding cell viability and metabolic activity (alamar blue assay) and how CBD influences the bone differentiation of exposed stem cells, by following the expression of the genes involved in osteogenesis as well as the mineralization process (alizarin red).

#### 2.4.1. Alamar Blue Cell Viability Assay

Alamar Blue is a reagent used to detect cell viability, but additionally can be used to check cell adhesion and proliferation. The rezaurin contained by Alamar Blue is a color indicator that has the ability to penetrate the cell membrane and get inside the metabolically active cells where it gets involved in a reduction process, which it transforms into resorufin. Resorufin is a red fluorescent dye of high intensity if there are many viable cells. The cells were seeded on 96 well plates at a cell density of 2 × 10^4^/well in 200 µL/well complete DMEM medium. Controls and treatments were done in triplicate. Treatments were performed at different doses of CBD (between 2 µM to 0.1 µM) and the viability was evaluated after 24, 96 h, and 8 days of culture in standard stem cell medium. The plates were incubated for 1 h at 37 °C, in the dark, with Alamar Blue in a dilution of 1/10 in complete medium. After the incubation, the fluorescence intensity was measured by using a BioTek Synergy 2 plate reader (excitation 540 nm, emission 620 nm). Cell viability and cell proliferation were evaluated in the same way after 24 h (1d), 3 days, and 8 days of the cultivation of stem cells. Statistical analysis was performed using a software GraphPad Prism 5.0, with “Two-way RM ANOVA” with Bonferroni posttest. A similar protocol was applied in an osteogenic differentiation experiment, dental derived stem cells being cultivated with OS and OC medium, and treated with different concentrations of CBD and Vit. D3. Alamar blue assay was done in day 12 of the differentiation experiment. Alizarin red was performed after 21 days of differentiation.

#### 2.4.2. Evaluation by RT-PCR of Genes Implicated in Osteogenic Differentiation

Dental pulp (DPSCs), apical papilla (APSCs), and dental follicle (DFSCs) cells were cultivated in OS medium in 6 well plates, with a seeding cell concentration of 2 × 10^5^ cells/well. In the second day of cultivation, cells were treated with Vit. D3 (2.5 nM) and CBD (0.75 µM). The treatments were repeated at each changing of medium (at 3 days). After 10 days of induction of differentiation, the cells were subject of RNA extraction, as described in Section 3.2. Sequences of primers used are illustrated in Table 2. Controls were cells cultivated in standard stem cell medium and in simple osteogenic medium, without treatments.

## 3. Results

Characterization of isolated cells

The isolated cells were propagated and at 6th passage, their characterization was performed by evaluating the expression of surface markers by flow cytometry and some genes characteristic of mesenchymal stem cells. The multipotent capacity was investigated by inducing differentiation into bone cells, chondrocytes, and adipocytes.

### 3.1. Flow Cytometry

The flow cytometry histograms are shown in Figure 1. We observed that DPSCs had an intense positive expression of mesenchymal stem cell markers: CD29 (92.2% of cells), CD49e (97.7%), CD73 (100%), CD90 (99.7%), CD105 (93.5%), CD166 (98%). DFSCs had a lower level of positive expression for CD29 (61.6%), but a high intensity for CD49e (87%), CD73 (98.7%), CD90 (98%), CD105 (81.6%), CD166 (93%). The APSCs had a lower expression of all adult mesenchymal stem cells markers: CD29 was expressed only in 1.4% of cells, CD49e (98.5%), CD73 (36.6%), CD90 (30.6%), CD105 (2.9%) and CD 166 (18.6%). All three cell types were negative for CD34, CD45, CD117, and HLA-DR.

### 3.2. RT-PCR-for Stemness Markers

The isolated cells were analyzed for some characteristic stem cells gene expression by RT-PCR: pou5F1 (Oct3/4), Nanog, and Vimentin. DFSCs express only oct3/4 gene at high levels. The APSCs cells showed high levels for Nanog. Only DPSCs cells expressed all three genes. (Figure 2).

### 3.3. Osteogenic Differentiation with Osteoinductive Medium

After the addition of a simple osteogenic differentiation medium, an increase in cell proliferation associated with morphological changes was observed and the cells became more elongated with dendritic extensions. Over time, the cellular layer was covered by amorphous deposits, suggesting the emergence of extracellular matrix synthesized de novo. Alizarin red staining performed after 28 days of cultivation showed the appearance of osteogenic nodules, more pronounced in the case of APSCs (Figure 3).

Immunocytochemical staining for some osteogenic markers revealed the strong expression for all molecules collagen 1A, osteopontin (OP), and osteocalcin (OC), with a higher intensity for DPSCs and DFSCs. Inducing osteogenic differentiation of APSCs caused a decrease in the number of cells and the expression of osteogenesis markers was less intense and not all cells expressed these markers. (Figure 4).

### 3.4. Chondroblastic Differentiation

Chondroblastic differentiation was evaluated by visualization of typical morphological changes of cells as well by Alcian blue staining on day 28 of differentiation. DFSCs and DPSCs showed a rapid cell aggregation as condrocytes spheroids, which increased in volume over time and were positive for Alcian blue, a stain for acidic polysaccharides such as glycosaminoglycans, some types of mucopolysaccharides, and sialylated glycocalyx. APSCs showed a much slower and less intense process of chondroblastic differentiation (Figure 5). 

### 3.5. Adipogenic Differentiation

Isolated cells were cultivated in adipogenic inducing medium for 14 days and stained with Red Oil, a lysochrome dyazo dye used for demonstrating neutral triglycerides and lipids in tissues. DFSCs and DPSCs cells presented lipid inclusions stained in red, almost in every cell. In contrast, APSCs cells did not gain this characteristic feature of adipocytes (Figure 6).

Influence of CBD treatment on osteogenic differentiation of dental derived stem cells

Working doses were established and the effect of CBD on cell proliferation and metabolism of cells cultured under standard conditions with stem cell culture medium was evaluated after 24 h, 4 days, and 8 days. Controls were untreated cells. Evaluation was performed with the Alamar Blue assay. In the first 24 h, no cytotoxicity was observed, even at a higher dose of 2.0 µM CBD. APSCs showed an increase of fluorescence at the lowest doses of CBD (0.25 µM and 0.1 µM), growth that we attributed to the increase in cellular metabolism rather than the increase in cell proliferation (Figure 7).

After 4 days of cultivation in the presence of CBD, DFSCs showed similar proliferation rates with control cells, even for the highest doses of CBD and a decreased cell proliferation at lower doses. An increased cell proliferation was observed for DPSCs and APSCs treated with the lowest dose of CBD (Figure 8).

The behavior of cells was different for the three cell types after eight days of cultivation in standard conditions and treatment with CBD. DFSCs doubled the number of cells and CBD induced a slight decrease in proliferation rate. DPSCs had a similar cell growth of treated cells with control cells. A stimulation of proliferation was observed for 0.25 µM CBD dose. APSCs had the lowest proliferation rate, but CBD treatment did not influence cell multiplication (Figure 9).

Another aim of this paper was to study the effects of CBD on the process of bone differentiation of MSCs isolated from dental tissues. The effects of CBD on stem cells were compared to untreated cells and those treated with the biological active form of vitamin D, 1α,25-dihydroxyvitamin D3 (1,25(OH)2D3, as well the combined influence of osteoinductive medium (OS medium and BMP2 supplemented OC medium). The cell viability and proliferation of cells cultivated with OC medium and treated with Vit. D3 (20 nM and 10 nM), and CBD (1 µM, 0.75 µM, 0.5 µM, and 0.25 µM) were investigated. Alamar blue test was performed after 12 days of cultivation. DFSCs, DPSCs, and APSCs reacted with the higher doses of CBD by decreasing cell number, a sign of possible induced cell differentiation. DPSCs showed a slow increase of fluorescence values for Vit. D3 treatments (Figure 10).

The mineralization process was investigated by staining of samples with Alizarin Red after 21 days of cultivation, comparing the effects of osteogenic medium (OS vs. OC), combined with the treatment with CBD (0.75 µM, 0.5 µM) and Vit. D3 (10 nM and 5 nM). Controls were DFSCs, APSCs, and DPSCs cultivated only with OS or with OC medium. DFSCs showed a spontaneous differentiation and associate mineralization increased rate induced only by the OS medium. The OC medium alone did not lead to the same effect observed in the OS medium, but treatments with 0.75 µM CBD and 10 nM Vit. D3 induced a slight increase in alizarin red levels. DPSCs did not respond different to treatments in comparison with the control cells, both at the OS and OC medium. The mineralization rate of treated APSCs (CBD and Vit. D3) was similar to that of the control cells in the case of the OS medium. The OC medium induced the best mineralization for APSCs alone or in combination with Vit. D3. (Figure 11).

RT-PCR was another method of investigating the bone differentiation of isolated cells, cultivated for 10 days in the presence of a standard stem cell medium, OS medium, and treatments with Vit. D3 (2.5 nM) and CBD (0.75 µM). The expression of early genes induced in the osteogenesis process (osteopontin (OP) and collagen 1A) and the genes involved later in this process (osteonectin and osteocalcin) was determined quantitatively. Each cell type behaved differently. DFSCs showed a spontaneous osteogenic differentiation only with standard stem cell medium with the highest expression for all genes. DPSCs cells were most responsive to Vit. D3 treatment with increased expression for OP, collagen1, and osteocalcin, but also to CBD treatment that induced the expression of osteocalcin gene. APSCs cells responded the best to CBD treatment with high RNA levels for OP, osteonectin, and osteocalcin. The expression of collagen 1 and in a lesser level, OP and osteonectin, was induced by Vit. D3 treatment (Figure 12).

## 4. Discussion

The aim of our paper was to stimulate DPSCs, DFSCs, and APSCs to differentiate into osteoblasts using CBD and to compare the results with a standard osteoblast enhancer, Vit. D3. All stem cell types used in our study are easy to harvest from dental tissues of an impacted third molar during or after odontectomy. Due to the malposition and impossibility to erupt, many impacted third molars are extracted and their tissues, otherwise lost, can be used in therapeutic purposes, like stem cell harvesting. The main advantage of using stem cells for bone regeneration is the use of autologous cells, avoiding some of the risks allografts come with. It is well known that oral tissues are abundant in mesenchymal stem cells [33] and that differentiation into at least three cell lines is one characteristic that validates their stemness capacity. Dental tissue derived MSCs are considered the perfect candidates as an accessible source of high-quality stem cells to be used in bone regeneration, given their already proven ability of osteogenesis, osteoinduction and osteoconduction [34]. The isolated cells from the dental tissues (DFSCs, DPSCs and APSCs) showed mesenchymal stem cell characteristics but different profiles were outlined for each cell type. The most typical profile was found at DPSCs, 90–100% of cells being positive for CD29, CD49e, CD73, CD90, CD105, and expressed the key regulatory genes in maintenance of pluripotency and self-renewal of Oct3/4 and Nanog and the mesenchymal characteristic gene, vimentin. DPSCs showed multipotent differentiation into osteoblasts, chondrocytes, and adipocytes. DFSCs had a lower level of positive expression for CD29 but a high intensity for CD49e, CD73, CD90, CD105, CD166, and the genetic profile showed high expression for Oct3/4 gene. DFSCs differentiated into bone cells with the best mineralization level, chondroblasts, and adipocytes. Both cell types DPSCs and DFSCs had a high proliferation rate, which was maintained at more advanced passages. Cells isolated from apical papilla showed a more heterogenous cell population as flow cytometry revealed low positivity for CD29 and CD105 and 18–30% of cells’ positivity for CD73, CD90, CD105, and CD 166; thus, it did not meet the minimum criteria described by Dominici et al. [35]. It seems to be a heterogeneous population that can include especially perivascular cells (which in turn may have characters of stemness). APSCs expressed in high levels only CD49e (α5 subunit of β51 integrin subunit of the fibronectin receptor), a surface molecule implicated in interaction with extracellular matrix (ECM), with role in signaling between the mesoderm and the neural crest, thereby regulating neural crest-dependent morphogenesis of essential embryonic structures [36]. Of the cell types used, the DPSCs were the only cells which expressed all three genes we analyzed. This suggests the superior stemness capacity of the DPSCs. DFSCs and APSCs each expressed a different gene of those analyzed, showing that even though we have chosen cell types only from soft tissues, they are still different from each other. This was also shown in the differentiation process. The three stem cell types, APSCs to a lesser extent, used in our study were able to differentiate into osteoblasts, chondroblasts, and adipocytes and therefore they have proven another stemness characteristic. DPSCs and DFSCs have shown a superior osteogenic and chondrogenic and adipogenic differentiation capacity than APSCs.

Dental pulp is an intense innervated and vascularized tissue, found inside the crown and the root canals, which contains various types of cells like stem cells, fibroblasts, and odontoblasts [37]. DPSCs have originated in mesodermal tissues [38] and have fibroblast-like, elongated, spindle-shaped morphology [39]. Their progenitors have not yet been identified [40]. DPSCs are proven to have the ability to differentiate into various types of cells as mesenchymal cells: chondrocytes, cardiomyocytes, adipose cells, muscle cells, osteoblasts and odontoblasts, and nonmesenchymal cells as neuronal cells and melanocytes [39]. They have a high proliferation potential [41]; therefore, they were considered suitable for being used in tissue regeneration [42]. The dental follicle is an ectomesenchymal derivate that circumscribes the tooth during eruption and is involved in the tooth development process through osteoclast management and bone formation. When the tooth is erupted, the dental follicle ends up transforming into periodontium. DFSCs have fibroblast-like morphology and can differentiate into osteoblasts, adipocytes, neurons, and cementoblasts [40]. DFSCs have an increased proliferation and immunomodulatory potential [41]. The in vitro results of comparative studies suggest that DPSCs have a higher capacity of producing hard tissue than DFSCs [43]. The apical papilla is a tissue localized on the end of the tooth root during tooth development [44]. It is involved into the tooth development process by differentiating into the dental pulp. They have also proven their multilineage differentiation into osteoblasts, chondroblasts, and neurons, and have a high proliferation rate [40]. In our study, the cells isolated from the apical papilla had an ambiguous phenotype of mesenchymal stem cells and are differentiated only into the bone and chondroblastic lineage and not into the adipocytes.

There are successfully used several mechanisms of inducing osteogenic differentiation of DPSCs, like gene expression inhibition [45], identification of circular RNA mechanisms, which are recognized to have a crucial role in bone regeneration correlated with stem cells [46]. Okajacekova et al. [47] tested the osteoblast differentiation of DPSCs by keeping them under different conditions and found that only one osteogenic medium xeno-free without animal serum supplement (OsteoMAX-XF™ Differentiation Medium) showed a rapid and strong calcification [47]. DFSCs and APSCs have also proven their osteoblast differentiation potential in different conditions, as bone regeneration on titanium implants [48] or cranial defects [49], for DFSCs or on 3D scaffolds [50] and Silk Fibroin Scaffolds [51] for APSC.

All these diverse methods of stimulating and testing the osteoblast differentiation of MSCs derived from dental tissues and many other methods are successful. However, we explored cannabinoids as a potential osteogenic enhancer of the process of bone regeneration.

Treatments of isolated stem cells with CBD and Vit. D3 in standard conditions for 8 days and in complex osteoinductive medium with BMP-2 for 12 days showed no cytotoxicity with CBD. In the osteogenic experiments DFSCs, DPSCs, and APSCs reacted to the higher doses of CBD by decreasing cell number, a sign of possible induced cell differentiation. The influence of CBD and Vit. D3 on bone differentiation was investigated using two kinds of osteinductive mediums (simple osteogenic medium and complex osteogenic medium supplemented with BMP-2). The differences of cell response to CBD (expression of osteopontin, collagen 1A, osteonectin, and osteocalcin genes) and mineralization (alizarin red) can be explained as the first intention by the expression of CB2R on isolated cells. APSCs were shown to be the most responsive to CBD in inducing bone protein expression but not in mineralization. The cells’ response to osteogenic stimuli in combination with CBD or Vit. D3 was different between cell types in terms of expression of bone specific genes (osteopontin, collagen 1A, osteonectin, and osteocalcin) and mineralization. DFSCs showed a spontaneous bone differentiation in standard stem cell medium and simple osteogenic medium. DPSCs had the best osteogenic response to Vit. D3 (2.5 nM) as gene expression but with the lowest mineralization rate (regardless of cultivation conditions). CBD in low dose (0.75 µM) induced osteogenic gene expression and Vit. D3 determined an increased mineralization rate of APSCs cells.

Immediate clinical applications of CBD administration would be the use of CBD as an adjuvant in bone fractures induced by metastases in which the analgesic effect could be combined with the osteoinductive one. Osteoporosis is also accompanied by chronic, unbearable pain; this would be another possibility to treat both chronic pain and osteoporosis. Combination of cellular therapies and targeting the CB system could be an alternative therapeutic solution in complex complicated fractures. CBD could open a new era in bone regeneration and osseointegration enhancement. Besides CBD is already known to improve bone regeneration [27], it was also proven to increase mesenchymal stem cell migration [52], which leads to faster wound healing. In addition, CBD reduces inflammation [53] and has anti-tumoral [54] and antioxidant [55] properties, which can increase the survival rate of stem cells after they are placed in a receptor site. Bone regeneration using CBD and Vit. D3 could also be effective for treating periodontitis. Periodontitis can be detected early by evaluating immunoglobulin antibodies [56,57], but in its advanced stages, if not treated, it can have severe irreversible consequences. Several treatments have been proposed for periodontitis [58], but a standard treatment that can successfully be applied in all clinical cases has not been discovered yet.

We did not find in the literature studies analyzing dental tissue derived stem cells differentiating into osteoblasts, stimulated by cannabis; hence, we were unable to make a comparison with other results.

All these results lead to our perspective of stimulating the osteogenic differentiation of dental MSCs using CBD.

## 5. Conclusions

The data presented above shows that mesenchymal stem cells isolated from dental tissues are capable of differentiating into osteoblasts under various conditions. DFSCs, DPSCs, and APSCs respond differently to the same osteogenic stimuli. DPSCs revealed the best osteogenic potential when treated with Vit. D3, the lowest dose of CBD induced the best bone proteins expression for APSCs, and DFSCs showed the highest mineralization capacity for treatments with CBD and Vit. D3, implying that all stem cell lineages used in our study are suitable for bone regeneration.

This study provides evidence for a better understanding of the effects of CBD and Vit. D3 on the populations of MSCs with dental origin, supporting the development of tissue engineering in the dentistry field. More studies are needed in order to establish an osteogenic differentiation protocol using CBD and Vit. D3 on dental MSCs. 

## Figures and Tables

**Figure 1 medicina-56-00607-f001:**
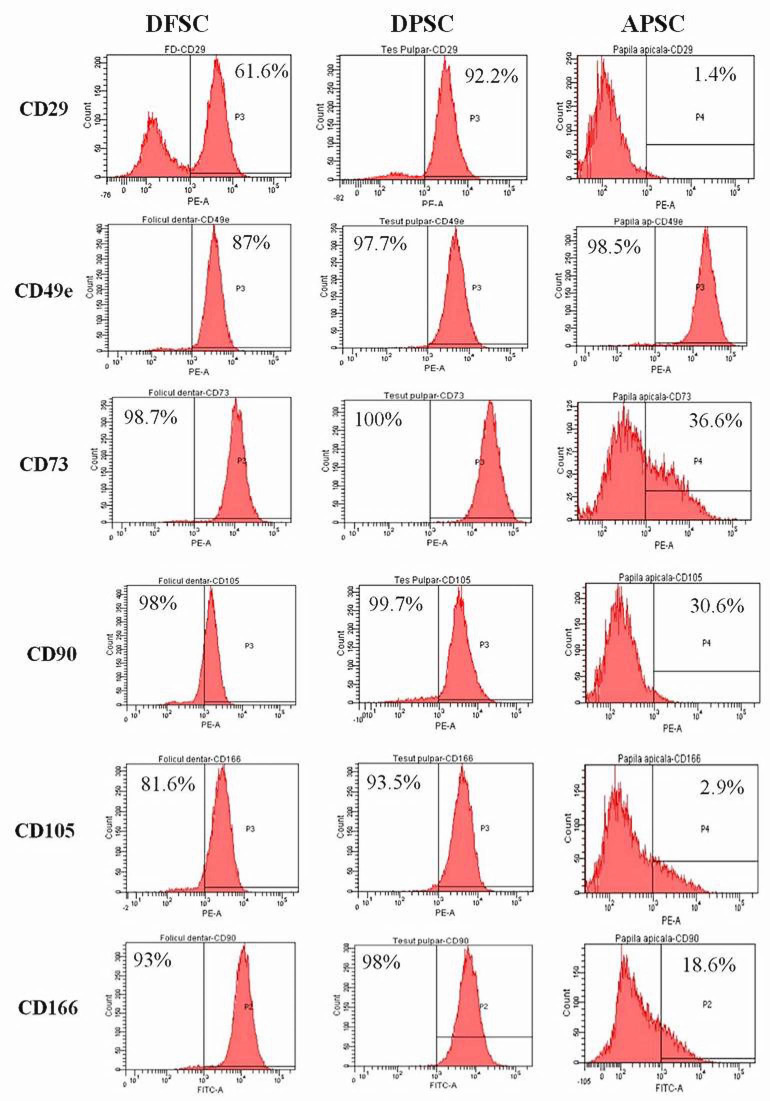
The three types of isolated cells from the dental follicle (DFSCs), dental pulp (DPSCs), and apical papilla (APSCs) analyzed by flow cytometry for surface stem cell markers.

**Figure 2 medicina-56-00607-f002:**
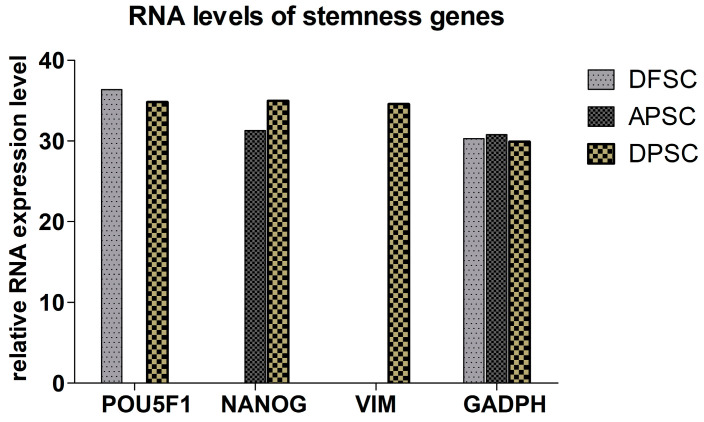
Quantitative RNA expression of genes implicated in osteogenic differentiation.

**Figure 3 medicina-56-00607-f003:**
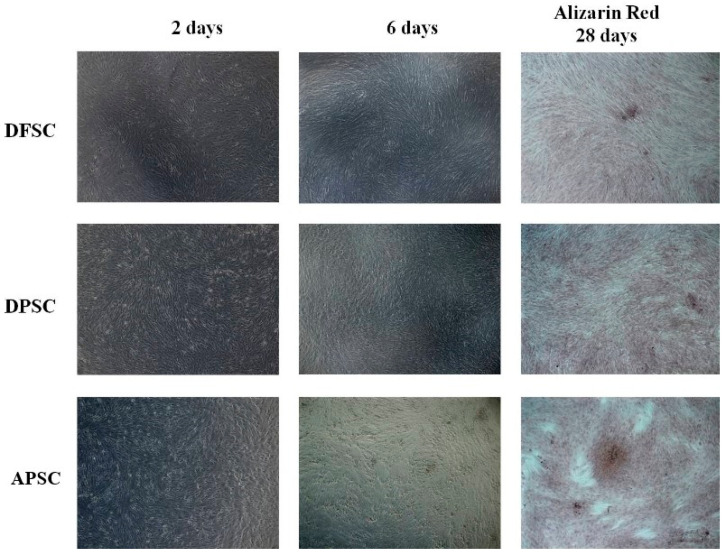
Osteogenic differentiation with osteoinductive medium. White light microscopy images of morphological changes induced by cultivation of cells with simple osteogenic medium (after 2 and 6 days). Alizarin red staining of differentiated cells after 28 days of cultivation.

**Figure 4 medicina-56-00607-f004:**
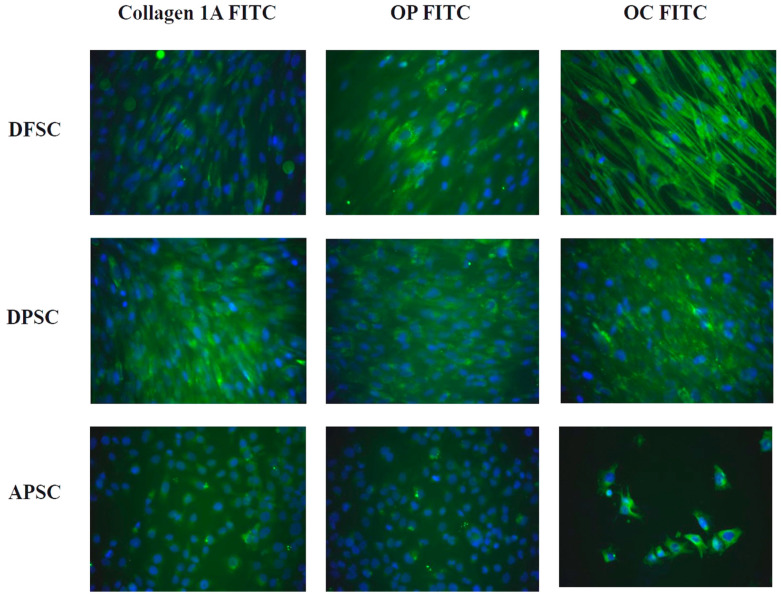
Osteogenic differentiation of dental follicle stem cells (DFSCs), dental pulp stem cells (DPSCs), and apical papilla stem cells (APSCs) stained with monoclonal antibodies for collagen 1A, osteopontin (OP), and osteocalcin (OC) assessed by immunofluorescence.

**Figure 5 medicina-56-00607-f005:**
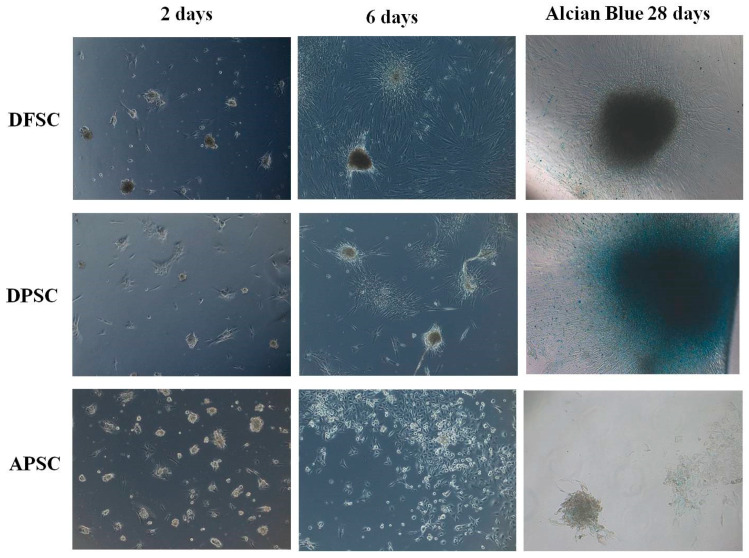
White light microscopy images of DFSCs, DPSCs, and APSCs induced for chondroblastic differentiation, on day 2, day 6, and day 28 (Alcian blue staining).

**Figure 6 medicina-56-00607-f006:**
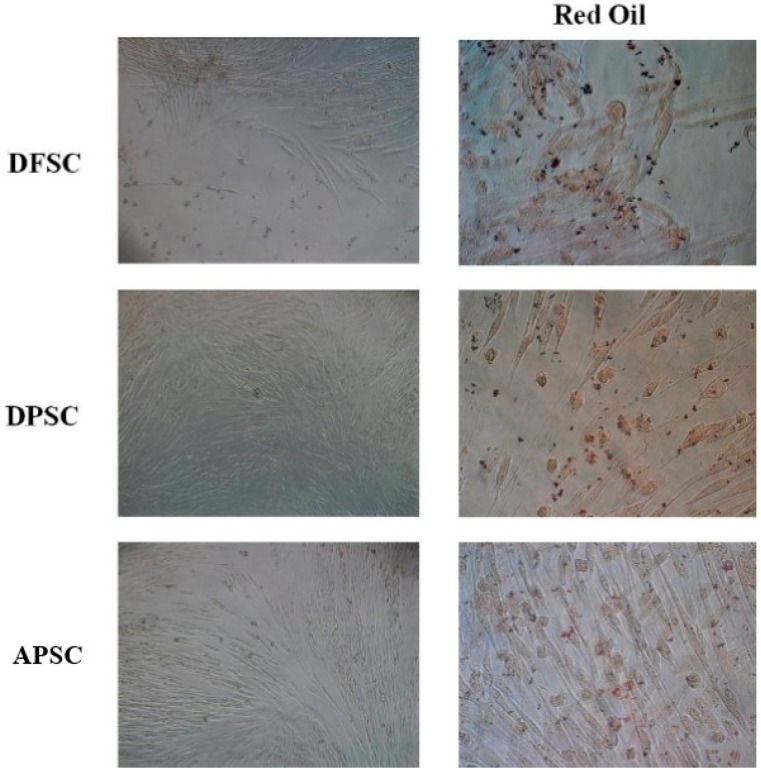
Images in light microscopy of induced DFSCs, DPSCs, and APSCs for adipogenic differentiation, cultivated for 14 days in adipogenic medium. Right panel: red oil stain.

**Figure 7 medicina-56-00607-f007:**
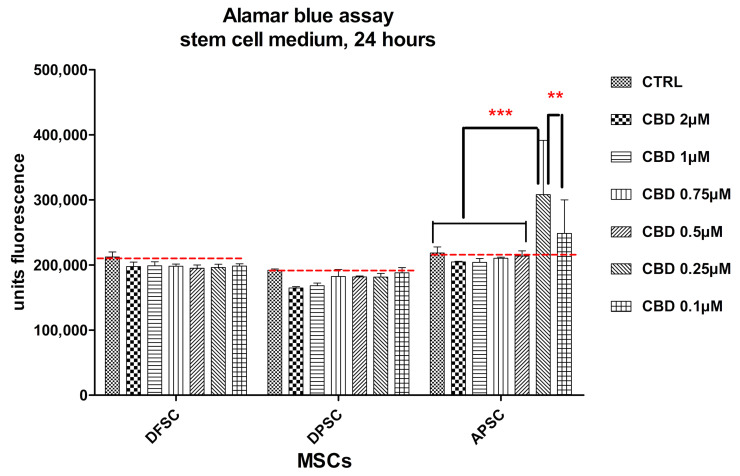
Graphical representation of alamar blue results. DFSCs, DPSCs, and APSCs viability evaluated after 24 h of treatment with different CBD doses. Cells were cultivated in standard stem cell medium. (**, *p* ≤ 0.01; ***, *p* ≤ 0.001)

**Figure 8 medicina-56-00607-f008:**
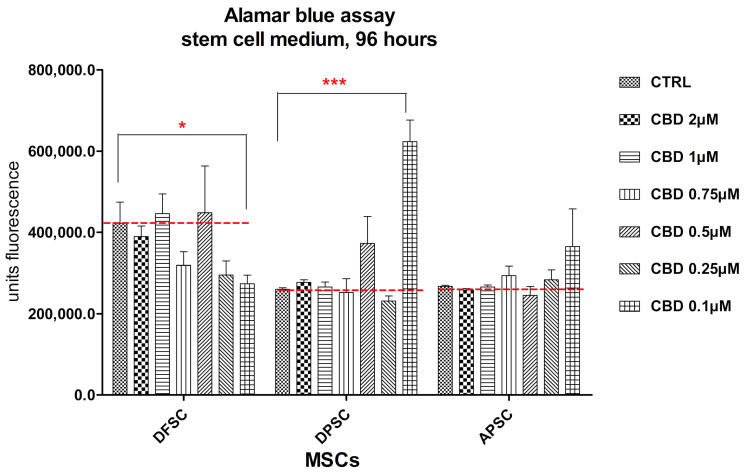
Graphical representation of alamar blue results. DFSCs, DPSCs, and APSCs viability evaluated after 96 h. (*, *p* ≤ 0.05; ***, *p* ≤ 0.001)

**Figure 9 medicina-56-00607-f009:**
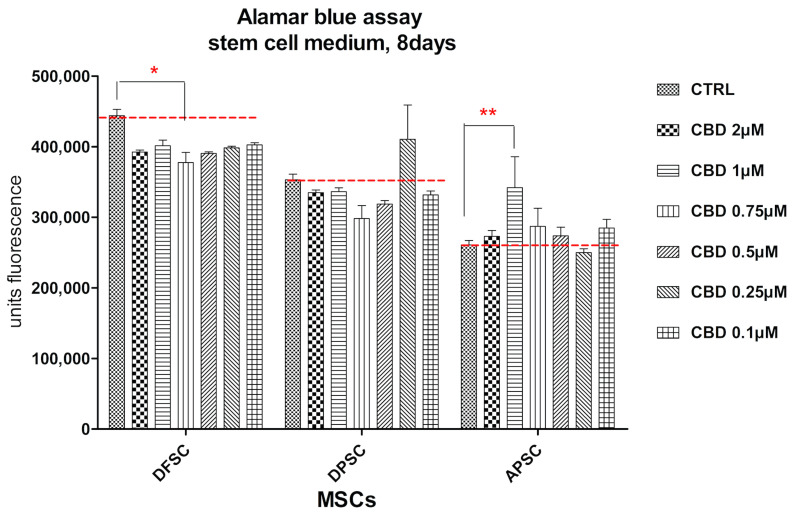
Graphical representation of alamar blue results. DFSCs, DPSCs, and APSCs viability evaluated after eight days. (*, *p* ≤ 0.05; **, *p* ≤ 0.01)

**Figure 10 medicina-56-00607-f010:**
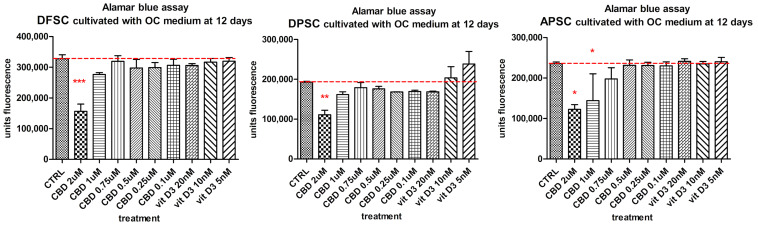
Alamar blue viability test of DFSCs, DPSCs, and APSCs treated with different doses of CBD and Vit. D3 cultivated 12 days in OC medium. (*, *p* ≤ 0.05; **, *p* ≤ 0.01; ***, *p* ≤ 0.001)

**Figure 11 medicina-56-00607-f011:**
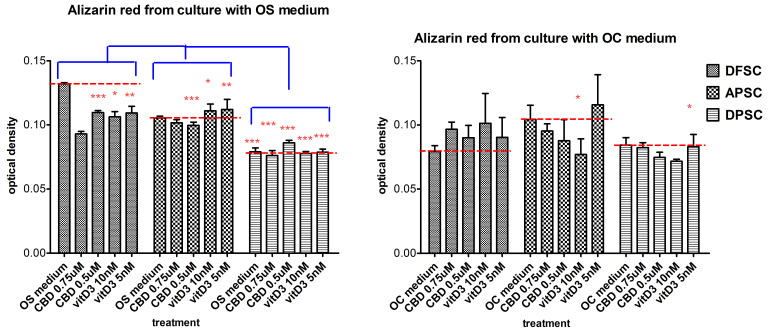
Graphical representation of alizarin red optical density levels of DFSCs, DPSCs, and APSCs cultivated with OS medium (**a**) and OC medium (**b**) for 21 days, and treated with CBD and Vit. D3. (*, *p* ≤ 0.05; **, *p* ≤ 0.01; ***, *p* ≤ 0.001).

**Figure 12 medicina-56-00607-f012:**
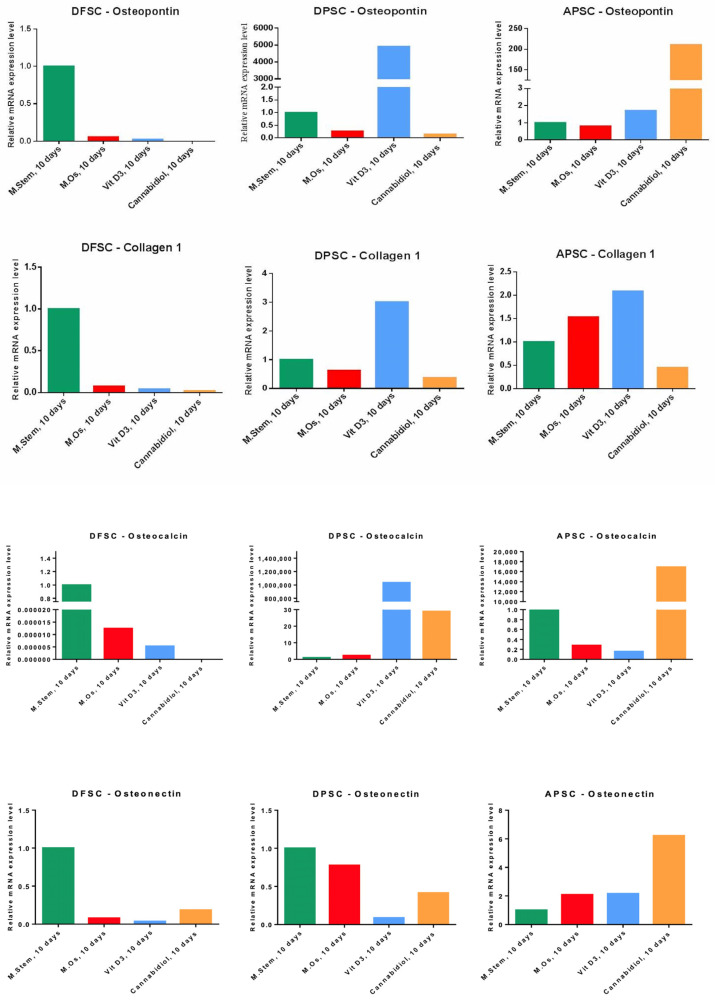
Graph representation of relative RNA expression levels of osteogenic genes (osteopontin, collagen 1, osteocalcin, and osteonectin) of DFSCs, DPSCs, and APSCs after 10 days of cultivation with standard stem cell medium, OS medium, and treatments with Vit. D3 (2.5 nM) and CBD (0.75 µM).

**Table 1 medicina-56-00607-t001:** Primer sequences for genes in stemness characterization.

NANOG left:	NANOG right:
5′ CAGTCTGGACACTGGCTGAA 3′	5′ CACGTGGTTTCCAAACAAGA 3′
ZFP42 left:	ZFP42 right:
5′ GGCCTTCACTCTAGTAGTGCTCA 3′	5′ CTCCAGGCAGTAGTGATCTGAGT 3′
VIM left:	VIM right:
5′ TGGTCTAACGGTTTCCCCTA 3′	5′ GACCTCGGAGCGAGAGTG 3′
POU5F1 left:	POU5F1 right:
5′ TGAGTAGTCCCTTCGCAAGC 3′	5′ GAGAAGGCGAAATCCGAAG 3′
GAPDH right (rw):	GAPDH left (fw):
5′ GCATGGACTGTGGTCTGCAA 3′	5′ GGACTGAGGCTCCCACCTTT 3′

**Table 2 medicina-56-00607-t002:** Primers’ sequences of the osteogenic genes used for RT-PCR analysis of stem cells differentiated and treated with cannabidiol and Vit. D3.

ACTB left: 5′ TCCAAATATGAGATGCGTTGTT 3′	ACTB right: 5′ TGCTATCACCTCCCCTGTGT 3′
GAPDH left (rw): 5′ GCATGGACTGTGGTCTGCAA 3′	GAPDH right (fw): 5′ GGACTGAGGCTCCCACCTTT 3′
Osteonectin Left: 5′ TCTTCCCTGTACACTGGCAGTTC 3′	Osteonectin Right: 5′ AGCTCGGTGTGGGAGAGGTA 3′
Colagen1 Left: 5′ GGGATTCCCTGGACCTAAAG 3′	Colagen1 Right: 5′ GGAACACCTCGCTCTCCA 3′
Osteopontin Left: 5′ CAGTGACCAGTTCATCAGATTCATC 3′	Osteopontin Right: 5′ CTAGGCATCACCTGTGCCATACC 3′
Osteocalcin Left: 5′ ATGAGAGCCCTCACACTCCT 3′	Osteocalcin Right: 5′ CAAGGGGAAGAGGAAAGAAG 3′

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
