# Peer review of "Cannabidiol and Vitamin D3 Impact on Osteogenic Differentiation of Human Dental Mesenchymal Stem Cells"

_medicina, 2020, doi:10.3390/medicina56110607_

Round 1

Reviewer 1 Report

The paper entitled “Cannabidiol and Vitamin D3 impact on osteogenic differentiation of human dental mesenchymal stem cells” investigates the potential of mesenchymal stem cells extracted from three different dental tissues (DPSC, DFSC and APSC) to differentiate towards osteogenic, chondrogenic and adipogenic lineages, as well as the effect of cannabidiol and vitamin D3 to induce their osteogenic differentiation in vitro.

This is an interesting topic and the manuscript contains significant amount of data. However, some aspects of the manuscript should be considered:

  1. The cells used in this study were extracted from a single patient. The authors should comment on the reproducibility of the cell behaviour observed in this study.
  2. The authors state: “Dental pulp contains various types of cells like stem cells, fibroblasts and odontoblasts”. How were the MSCs sorted in this study? It looks like flow cytometry was only used to characterise the cell populations. Cell isolation process should be clarified.
  3. The authors state: “Cells isolated from apical papilla showed a more heterogenous cell population as flow cytometry revealed with low positivity for CD29 and CD105 and 18-30% of cells` positivity for CD73, CD90, CD105 and CD 166, so did not meet the minimum criteria described by Dominici et al.” According to these results, why did the authors continue using these cells in this study?
  4. Authors mention that they “explored cannabinoids as a potential osteogenic enhancer, aiming to simplify the process of bone regeneration”. In which sense the use of cannabidiol would simplify the bone regeneration process?
  5. The authors state: “The differences of cell response to CBD (the expression of osteopontin, collagen 1A, osteonectin and osteocalcin genes) and mineralization (alizarin red) can be explained as the first intention by the expression of CB2R on isolated cells. Have they analysed this here?
  6. Scale bars should be added to all the figures containing microscopy images (Figures 3, 4, 5 and 6).
  7. The authors mention that cell morphology is evaluated; however, these results are not discussed in the manuscript. Also, the magnification used for microscopy images looks small to appreciate changes in cell morphology. Have this been quantified somehow?
  8. Have the authors quantified the Alizarin Red, Alcian Blue and Red oil staining results?
  9. Control samples used for each experiment should be clarified.
  10. How did the authors select the different time points for the differentiation studies? OCN is a late osteogenic marker. Why the gene expression for OCN was evaluated at 10 days?
  11. Figure 12 does not have error bars. How many replicates were used for this experiment?
  12. The authors mention that vitamin D3 was used as a standard osteogenesis promoter in order to measure the effects of CBD. Was vitamin D3 used as a positive control? This is not clear in the manuscript.
  13. According to the gene expression (in figure 12), it looks like the osteogenic controls don’t work as expected. The authors should comment on this. It looks like authors used vitamin D3 as a positive control to evaluate the effect of CBD. However, the use of vitamin D3 only induced the expression of some osteogenic markers.
  14. The effect of Vitamin D (described in the last paragraph of the discussion) should be explained earlier. The last paragraph of the discussion should summarise the findings of the paper instead.
  15. Please add reference for this statement in the introduction. “In dentistry, bone defects are caused by dental extractions, periodontal disease, peri-implantitis, apicoectomy, cystectomy, atrophy of the maxillary or mandibular alveolar bone, resection of the maxillary or mandibular bone, osteomyelitis”.
  16. The following sentence from the introduction is not clear and should be rephrased. It also needs a reference: “In bone regeneration are implicated cells (monocyte-macrophage-osteoclast lineage and the mesenchymal stem cell-osteoblast lineage), complex interactions between cells and microenvironments, all determining the activation of several pathways.”
  17. The manuscript should be proofread before publication. Some sentences should be rephrased.

As an example: “The stem cells were isolated and characterized using flowcytometry, reverse transcription polymerase chain reaction (RT-PCR), and osteogenic, chondrogenic and adipogenic differentiation”. You don’t “use” osteogenic differentiation.

Terms like “marker positiveness” or “Low positivity” should be rephrased.

“Osteodifferentiation of DPSCs” should be osteogenic differentiation of DPSCs.

Examples of some typos: Page 2 (abstract): “flowcytometry” should be “flow cytometry”. Line 85: remove “in”. Line 515: “corelated” should be “correlated”.

Author Response

            Thank you for allowing us to resubmit a revised version of our manuscript. Please accept our revised version for further consideration. We would like to express our gratitude for providing constructive feedback by identifying the areas of our manuscript that needed further improvements. We appreciate the tremendous effort and time you devoted to strengthen our manuscript. Accordingly, we have uploaded the revised manuscript. Please find below our response. We hope this will ease the reading of the paper and we are confident that the new version of the manuscript is significantly improved. Thus, we look forward to hearing from you and to respond to any other questions or comments you may have.

The paper entitled “Cannabidiol and Vitamin D3 impact on osteogenic differentiation of human dental mesenchymal stem cells” investigates the potential of mesenchymal stem cells extracted from three different dental tissues (DPSC, DFSC and APSC) to differentiate towards osteogenic, chondrogenic and adipogenic lineages, as well as the effect of cannabidiol and vitamin D3 to induce their osteogenic differentiation in vitro.

This is an interesting topic and the manuscript contains significant amount of data. However, some aspects of the manuscript should be considered:

Thank you for the rigorousness with which you analyzed the paper and the relevant comments.

  1. The cells used in this study were extracted from a single patient. The authors should comment on the reproducibility of the cell behaviour observed in this study.

This is a pertinent question. Indeed, the use of cells from a single patient is questionable, but from the experience of our laboratory, isolating stem cells from dental tissues from several patients of different ages, MSCs expressed markers of stemness quite similar, the only differences being regarding their proliferative capacity, in older patients this capacity is diminished. In other publications of our team we reported similar characters for cells derived from dental follicle, periodontal ligament, gingival tissue:

Ondine Lucaciu, Olga Soriţău , Dan Gheban, Dan Rus Ciuca, Oana Virtic , Adriana Vulpoi, Noemi Dirzu , Radu Câmpian , Grigore Băciuţ , Catalin Popa , Simion Simon , Petru Berce , Mihaela Băciuţ , Bogdan Crisan  Dental follicle stem cells in bone regeneration on titanium implants BMC Biotechnol. 2015 Dec 30;15:114. doi: 10.1186/s12896-015-0229-6.PMID: 26718927

Boșca AB, Ilea A, Sorițău O, Tatomir C, Miklášová N, Pârvu AE, Mihu CM, Melincovici CS, Fischer-Fodor E Modulatory effect of curcumin analogs on the activation of metalloproteinases in human periodontal stem cells..Eur J Oral Sci. 2019 Aug;127(4):304-312. doi: 10.1111/eos.12625. Epub 2019 Jul 3.PMID: 31270880

Soancă A, Lupse M, Moldovan M, Pall E, Cenariu M, Roman A, Tudoran O, Surlin P, Șorițău O.
Applications of inflammation-derived gingival stem cells for testing the biocompatibility of dental restorative biomaterials.Ann Anat. 2018 Jul;218:28-39. doi: 10.1016/j.aanat.2018.02.009. Epub 2018 Mar 28.PMID: 29604386

P, Hedesiu M, Soritau O, Perde-Schrepler M, Brie I, Pall E, Fischer-Fodor E, Bogdan L, Lucaciu O, Belmans N, Moreels M, Salmon B, Jacobs R. Low-dose radiations derived from cone-beam CT induce transient DNA damage and persistent inflammatory reactions in stem cells from deciduous teeth.

Dentomaxillofac Radiol. 2019 Jan;48(1):20170462. doi: 10.1259/dmfr.20170462. Epub 2018 Aug 31.PMID: 30168750

Páll E, Florea A, Soriţău O, Cenariu M, Petruţiu AS, Roman A. Comparative Assessment of Oral Mesenchymal Stem Cells Isolated from Healthy and Diseased Tissues.Microsc Microanal. 2015 Oct;21(5):1249-63. doi: 10.1017/S1431927615014749. Epub 2015 Aug 28.PMID: 26315895.

 As demonstrated in the present paper, the data obtained correlate well with others in the literature, especially in terms of dental follicle stem cells (DFSC) and dental pulp stem cells (DPSC). On the other hand, isolating cells from a single patient could show how stem cells of different origins contribute to the local regeneration of dental tissues with a same genetic signature. Also the fact that the experiments were many and complex was another reason why we used cells isolated from a single patient.

  1. The authors state: “Dental pulp contains various types of cells like stem cells, fibroblasts and odontoblasts”. How were the MSCs sorted in this study? It looks like flow cytometry was only used to characterise the cell populations. Cell isolation process should be clarified.

MSC cells were not sorted but obtained by the method applied all over the world, namely the adhesion to plastic surfaces. It is a known method that is in fact part of the definition of adult mesenchymal stem cells as developed by Dominici et al in 2006. Many laboratories work according to this method. Mesenchymal stem cells have a high tendency to adhere to plastic but also to proliferate, having a clear advantage in the first days of cultivation in the standard environment over other cell types present in different tissues.

M Dominici 1, K Le Blanc, I Mueller, I Slaper-Cortenbach, Fc Marini, Ds Krause, Rj Deans, A Keating, Dj Prockop, Em Horwitz Minimal criteria for defining multipotent mesenchymal stromal cells. The International Society for Cellular Therapy position statement Cytotherapy. 2006;8(4):315-7. doi:10.1080/14653240600855905.PMID: 16923606 (“First, MSC must be plastic-adherent when maintained in standard culture conditions. Second, MSC must express CD105, CD73 and CD90, and lack expression of CD45, CD34, CD14 or CD11b, CD79alpha or CD19 and HLA-DR surface molecules. Third, MSC must differentiate to osteoblasts, adipocytes and chondroblasts in vitro”.)

In present paper we mentioned in 2.2 Stem cell isolation section, page 5, line 198 : The cells were isolated using the explant technique described by Centeno.

  1. The authors state: “Cells isolated from apical papilla showed a more heterogenous cell population as flow cytometry revealed with low positivity for CD29 and CD105 and 18-30% of cells` positivity for CD73, CD90, CD105 and CD 166, so did not meet the minimum criteria described by Dominici et al.” According to these results, why did the authors continue using these cells in this study?

We continued to use those cells, because 18% of cells were positive for all stemness markers, and were negative for CD45,CD34 and HLA-DR, and because although it seems to be a heterogeneous population that can include especially perivascular cells (which in turn may have characters of stemness, even if they do not express characteristic markers described in 2006 by Dominici et al. ) as it appears from his work : Ola A. Nada1,2,* and Rania M. El Backly2,3 Stem Cells From the Apical Papilla (SCAP) as a Tool for Endogenous Tissue RegenerationFront Bioeng Biotechnol. 2018; 6: 103.Published online 2018 Jul 24.  doi: 10.3389/fbioe.2018.00103, PMCID: PMC6066565PMID: 30087893.

We made some cited text selection from this paper that support our hypothesis:

  • “Studies have characterized the phenotypic traits as well as other regenerative potentials of these cells. Specific sub-populations have been highlighted as well as their neurogenic and angiogenic properties. Nevertheless, in light of the previously discussed features and potential applications of SCAP, there is still much to understand and a lot of information to unravel. The current review will discuss the role of specific markers for detection of different functional populations of SCAP; including CD146 and STRO-1, as well as their true multilineage differentiation potential.”
  • “Different SCAP phenotypes have been suggested to correlate to distinct functional properties (Bakopoulou et al., 2012). Research has shown that MSCs can be found in vascular tissues (Lv et al., 2014). Reflecting the perivascular location of SCAP is their positive expression of stem cell (SC) markers such as STRO-1 and CD146, with both expressions fading with passaging (Bakopoulou et al., 2012; Schneider et al., 2014; Liu et al., 2016). Moreover, SCAP also positively express CD73, CD90, and CD105; typical MSC markers (Hilkens et al., 2017).
  • Finally, SCAP have been found to simultaneously minimally or negatively express CD14, CD31, CD34, CD45, and HLA DR (Huang et al., 2009; Cao et al., 2013; Schneider et al., 2014; Hilkens et al., 2017). The negative expression of the leukocyte precursor CD45 is particularly vital as it confirms the stromal origin of SCAP and the absence of hematopoietic precursor contamination (Sonoyama et al., 2006; Huang et al., 2009; Bakopoulou et al., 2011, 2012; Li et al., 2014). Therefore, for efficient stem cell (SC) characterization, besides the basic International Society for Cellular Therapy (ISCT) criteria of positively expressing a group of antigens and negatively express others. The former being CD105, CD73, and CD90, and the latter being CD45, CD34, CD14 or CD11b, CD79a or CD19, and HLA-DR. (Lv et al., 2014). Furthermore, SCAP can specifically be characterized by the additional expression of CD24 and SCAP properties can be enhanced/altered by sorting the cells in accordance to STRO-1 and CD146. For instance, for studies targeting bone regeneration potential, it is recommended to preselect populations that are double positive for CD146 and STRO-1 (Bakopoulou et al., 2013), whereas when neurogenic regeneration is targeted, STRO-1 positive populations should be preselected (Huang et al., 2008). “

We added a short explanation in the text, page 20, line 488-489.

  1. Authors mention that they “explored cannabinoids as a potential osteogenic enhancer, aiming to simplify the process of bone regeneration”. In which sense the use of cannabidiol would simplify the bone regeneration process?

You are right, the term to simplify the process of bone differentiation is not appropriate. The differentiation process certainly cannot be simplified but it can be stimulated to be more efficient, which we have followed in this paper using CBD. We corrected in the paper this phrase.

  1. The authors state: “The differences of cell response to CBD (the expression of osteopontin, collagen 1A, osteonectin and osteocalcin genes) and mineralization (alizarin red) can be explained as the first intention by the expression of CB2R on isolated cells. Have they analysed this here?

This is the hypothesis we formulated, but for logistical reasons we have not yet been able to obtain the necessary reagents to prove it. This was mentioned as a suggestion for further research on the topic, as the authors could not find any study that evaluates the expression of CB2R on dental stem cells.

  1. Scale bars should be added to all the figures containing microscopy images (Figures 3, 4, 5 and 6).

The COVID-19 situation limits our acces to the laboratory equipped with the software in which we could make these changes. We will be able to do the changes after November 12, if considered absolutely necessary.

  1. The authors mention that cell morphology is evaluated; however, these results are not discussed in the manuscript. Also, the magnification used for microscopy images looks small to appreciate changes in cell morphology. Have this been quantified somehow?

The changes in cell morphology were evaluated in differentiation experiments. We have not quantified. It was followed only as a qualitative indicator of terminal differentiation process.

  • in osteogenesis (mentioned in the text at page 11, line 355-358, Fig. 3, for immunocytochemical staining- where in Fig.4 these changes are more easily visible even if the image size is small)
  • for chondroblastic differentiation page 13, line 376-378)

  1. Have the authors quantified the Alizarin Red, Alcian Blue and Red oil staining results?

For Alizarin red staining of treated samples with CBD and VitD3, as well as for the controls, quantification was performed after 21 days of cultivation with OS and OC medium. The resuls are illustrated in Fig. 11. We performed also quantification for Alcian blue and Red Oil, but because there were still many results in the paper, we considered that the images in fig. 3 and fig.5 are more eloquent.

  1. Control samples used for each experiment should be clarified.

We specified in the text for the experiments with alamar blue and alizarin red which are the control samples.

  1. How did the authors select the different time points for the differentiation studies? OCN is a late osteogenic marker. Why the gene expression for OCN was evaluated at 10 days?

It seems that OCN-known as a late bone marker, is present in lower concentration in early stages of osteogenic differentiation. “Osteocalcin plays a major role during the development of mesenchymal cells to osteoblasts and mature osteoid. It is assumed that during the early stage of osteoblast differentiation, low concentrations of osteocalcin promote the maturation of mesenchymal precursor cells into osteoblasts. At the end of the mineralization, high osteocalcin levels inhibit a prolonged ossification and formation of mature osteocytes.” [Banerjee C, et al. Differential regulation of two principal Runx2/Cbfa-1 n-terminal isoforms in response to bone morphogenetic protein-2 during development of the osteoblast phenotype. Endocrinology 2001;142:4026]

Alexander E Handschin 1, Marcus Egermann, Otmar Trentz, Guido A Wanner, Hans-Jürgen Kock, Gregor Zünd, Omana Anna Trentz Cbfa-1 (Runx-2) and osteocalcin expression by human osteoblasts in heparin osteoporosis in vitro Clin Appl Thromb Hemost. 2006 Oct;12(4):465-72. doi: 10.1177/1076029606293433.

  1. Figure 12 does not have error bars. How many replicates were used for this experiment?

In Figure 12 is the representation of relative expression of genes of interest (bone markers). The samples were in duplicate and in graphical representation are the average of the results.

  1. The authors mention that vitamin D3 was used as a standard osteogenesis promoter in order to measure the effects of CBD. Was vitamin D3 used as a positive control? This is not clear in the manuscript.

Yes, we used vitamin D3 as a positive control, as other studies already confirmed that it enhances osteogenic differentiation of stem cells. But it seems that not for all cells can be used as a positive control, it was not clearly mentioned in the text, maybe just in the discussion section, page 19 line 467-468.

  1. According to the gene expression (in figure 12), it looks like the osteogenic controls don’t work as expected. The authors should comment on this. It looks like authors used vitamin D3 as a positive control to evaluate the effect of CBD. However, the use of vitamin D3 only induced the expression of some osteogenic markers.

In our study, the intention was to use vitamin D3 as a positive control, but only DPSCs response clearly to vit. D3. Some reasons can be found regarding the stage in which the isolated cells are, DPSCs being the most undifferentiated. Another reason could be the expression of nuclear vitamin D receptors (VDR), which can be expressed differently by the 3 cell types, but also by the expression of their co-activators.

Posa et al (2018) demonstrated in a study with MSCs isolated from dental bud, that 1,25(OH)2D3 action is directed by inducing the expression of the typical osteoblastic markers. The action was particularly evident in the early stages of differentiation, decreasing over time. They concluded that Vit D acts on MSCs, driving the early phases of cell commitment toward the osteoblastic lineage. [Francesca Posa 1 2, Adriana Di Benedetto 1, Elisabetta A Cavalcanti-Adam 2, Graziana Colaianni 3, Chiara Porro 1, Teresa Trotta 1, Giacomina Brunetti 4, Lorenzo Lo Muzio 1, Maria Grano 3, Giorgio Mori 1 Vitamin D Promotes MSC Osteogenic Differentiation Stimulating Cell Adhesion and α V β 3 Expression Stem Cells Int. 2018 Feb 28;2018:6958713.

 doi: 10.1155/2018/6958713. eCollection 2018.PMID: 29681950]

Sylvia Christakos, Puneet Dhawan, Annemieke Verstuyf, Lieve Verlinden, and Geert Carmeliet Vitamin D: Metabolism, Molecular Mechanism of Action, and Pleiotropic EffectsPhysiol Rev. 2016 Jan; 96(1): 365–408.Published online2015 Dec 16.  doi: 10.1152/physrev.00014.2015PMCID: PMC4839493PMID: 26681795]

“The genomic mechanism of 1,25(OH)2D3 action involves the direct binding of the 1,25(OH)2D3 activated vitamin D receptor/retinoic X receptor (VDR/RXR) heterodimeric complex to specific DNA sequences. Numerous VDR co-regulatory proteins have been identified”. 

  1. The effect of Vitamin D (described in the last paragraph of the discussion) should be explained earlier. The last paragraph of the discussion should summarise the findings of the paper instead.

We have moved the paragraph to the introduction (and rearranged the following references).

  1. Please add reference for this statement in the introduction. “In dentistry, bone defects are caused by dental extractions, periodontal disease, peri-implantitis, apicoectomy, cystectomy, atrophy of the maxillary or mandibular alveolar bone, resection of the maxillary or mandibular bone, osteomyelitis”.

We have added reference (and rearranged the references).

  1. The following sentence from the introduction is not clear and should be rephrased. It also needs a reference: “In bone regeneration are implicated cells (monocyte-macrophage-osteoclast lineage and the mesenchymal stem cell-osteoblast lineage), complex interactions between cells and microenvironments, all determining the activation of several pathways.”

Indeed, we have rephrased. The reference it was already in the text: 14.  Emmanuel Gibon , Laura Y Lu , Karthik Nathan , Stuart B Goodman  Inflammation, ageing, and bone regeneration J Orthop Translat . 2017 Jul;10:28-35.  doi: 10.1016/j.jot.2017.04.002. Epub 2017 May 15. PMID: 29094003

  1. The manuscript should be proofread before publication. Some sentences should be rephrased.

As an example: “The stem cells were isolated and characterized using flowcytometry, reverse transcription polymerase chain reaction (RT-PCR), and osteogenic, chondrogenic and adipogenic differentiation”. You don’t “use” osteogenic differentiation.

Terms like “marker positiveness” or “Low positivity” should be rephrased.

“Osteodifferentiation of DPSCs” should be osteogenic differentiation of DPSCs.

Examples of some typos: Page 2 (abstract): “flowcytometry” should be “flow cytometry”. Line 85: remove “in”. Line 515: “corelated” should be “correlated”.

We have made the required changes.

Reviewer 2 Report

My main comment is base on the biological replicates. In the paper they use only one donor, that regarding the base of experimental approach and donor variation can affect all the result and conclusion. From my side the work has some news approach, but overall the main lack is the number of replicates. From my side these are more a preliminary data than data for a paper.

Author Response

            Thank you for allowing us to resubmit a revised version of our manuscript. Please accept our revised version for further consideration. We would like to express our gratitude for providing constructive feedback by identifying the areas of our manuscript that needed further improvements. We appreciate the tremendous effort and time you devoted to strengthen our manuscript. Accordingly, we have uploaded the revised manuscript. Please find below our response. We hope this will ease the reading of the paper and we are confident that the new version of the manuscript is significantly improved. Thus, we look forward to hearing from you and to respond to any other questions or comments you may have.

  1. My main comment is base on the biological replicates. In the paper they use only one donor, that regarding the base of experimental approach and donor variation can affect all the result and conclusion. From my side the work has some news approach, but overall the main lack is the number of replicates. From my side these are more a preliminary data than data for a paper.

This is a pertinent question. Indeed, the use of cells from a single patient is questionable, but from the experience of our laboratory, isolating stem cells from dental tissues from several patients of different ages, MSCs expressed markers of stemness quite similar, the only differences being regarding their proliferative capacity, in older patients this capacity is diminished. In other publications of our team we reported similar characters for cells derived from dental follicle, periodontal ligament, gingival tissue.

 As demonstrated in the present paper, the data obtained correlates well with others in the literature, especially in terms of dental follicle stem cells (DFSC) and dental pulp stem cells (DPSCs). On the other hand, isolating cells from a single patient could show how stem cells of different origins contribute to the local regeneration of dental tissues with a same genetic signature. Also, the fact that the experiments were many and complex was another reason why we used cells isolated from a single patient.

Although we could not argue all the results, for example, the hypothesis that CB1R expression would be the reason why the 3 cell populations responded differently to CBD, the results may guide future research (of our team or others), an aspect that we highlighted at the end of the manuscript.

Our work is honest, thanks for the comments will be helpful in the future.

Reviewer 3 Report

The paper entitled "Cannabidiol and Vitamin D3 impact on osteogenic differentiation of human dental mesenchymal stem cells" presents in a detailed and clear way effects of CBD on osteogenic differentiation of stem cells isolated from different dental origin. The manuscript is composed by different analysis  covering morphological, molecular and vitality fields.

However, I have some doubts about methodologies and discussion section:

  1. How is it possible to obtain osteogenic differentiation of stem cells seeded on 96 well culture plates? In my experience, mesenchymal stem cells need more space to stretch and differentiate in order to obtain bone nodules. In addition, it is very difficult to take a magnification of a 96 well. Please clarify.
  2. Alizarin red test: in figure 3 authors show Alizarin red staining of differentiated cells after 28 days. Why the bone nodules and the deposits are not marked red?
  3. Probably I miss the real purpose of the manuscript. What could be the real clinical applications? Which would be the most suitable cellular model from cranial bone tissue engineering in your experience? What is the advantage of CBD with respect to the normal osteogenic differentiation protocol?Please clarify in the discussion section.

Author Response

            Thank you for allowing us to resubmit a revised version of our manuscript. Please accept our revised version for further consideration. We would like to express our gratitude for providing constructive feedback by identifying the areas of our manuscript that needed further improvements. We appreciate the tremendous effort and time you devoted to strenghten our manuscript. Accordingly, we have uploaded the revised manuscript. Please find below our response. We hope this will ease the reading of the paper and we are confident that the new version of the manuscript is significantly improved. Thus, we look forward to hearing from you and to respond to any other questions or comments you may have.

The paper entitled "Cannabidiol and Vitamin D3 impact on osteogenic differentiation of human dental mesenchymal stem cells" presents in a detailed and clear way effects of CBD on osteogenic differentiation of stem cells isolated from different dental origin. The manuscript is composed by different analysis covering morphological, molecular and vitality fields.

Thank you for your pertinent observation.

However, I have some doubts about methodologies and discussion section:

  1. How is it possible to obtain osteogenic differentiation of stem cells seeded on 96 well culture plates? In my experience, mesenchymal stem cells need more space to stretch and differentiate in order to obtain bone nodules. In addition, it is very difficult to take a magnification of a 96 well. Please clarify.

Thank you for this observation, because you have drawn us attention to a mistake in manuscript, which we corrected. For differentiation experiments we cultivated the cells in 12 well plates. For alizarin red quantification we took aliquots in 96 well plates. The images from figure 3 were captured on 12 well plates.

  1. Alizarin red test: in figure 3 authors show Alizarin red staining of differentiated cells after 28 days. Why the bone nodules and the deposits are not marked red?

In figure 3 only in the right panel Alizarin red staining was performed, after 28 days of cultivation. Bone nodules in those images are present, maybe the resolution of images is not the best. The left and the middle panel (2 and 6 days) are unstained samples, there are only images that reflect the morphological changes observed in the early differentiation process.

  1. Probably I miss the real purpose of the manuscript. What could be the real clinical applications? Which would be the most suitable cellular model from cranial bone tissue engineering in your experience? What is the advantage of CBD with respect to the normal osteogenic differentiation protocol? Please clarify in the discussion section.

Clinical applications

  • A possible clinical application could be the control of oxidative stress may prevent and alleviate oral mucositis. Studies have demonstrated that cannabidiol is safe to use and possesses antioxidant, anti-inflammatory and analgesic properties. [L F Cuba1 2, F G Salum 1, K Cherubini 1, M A Z Figueiredo 1 Cannabidiol: an alternative therapeutic agent for oral mucositis? J Clin Pharm Ther . 2017 Jun;42(3):245-250. doi: 10.1111/jcpt.12504. Epub 2017 Feb 12.PMID: 28191662]
  • One of the possible clinical uses would be for example in bone fractures, bone induced by metastases in which the analgesic effect could be combined with the osteoinductive one. Osteoporosis is also accompanied by chronic, unbearable pain, here would be another possibility to treat both chronic pain and osteoporosis.

The most suitable cellular model is hard to say, it also depends on the cell source provided at a given time from a patient (dental pulp, dental follicle, apical papilla, gingival tissue and so one). The dental follicle regresses over time, so it will be found especially in young adults. As our work reflected, DFSCs are not so dependent from vitamin D, they differentiated spontaneously even without osteoinductive medium. DPSCs were most responsive to vitamin D and APSCs to CBD. The advantage of CBD with respect of normal osteogenic differentiation medium was observed especially for APSCs, so for these cells is indicated CBD.

Besides CBDs’ already known potential to improve bone regeneration (1), it was also proven to increase mesenchymal stem cells migration (2), which leads to a faster wound healing. In addition, CBD reduces inflammation (3) and has anti-tumoral (4) and antioxidant (5) proprieties, which can increase the survival rate of stem cells after they were placed into a receptor site.

  1. Amir Kamali 1 , Ahmad Oryan 2 , Samaneh Hosseini  3 , Mohammad Hossein Ghanian  4 , Maryam Alizadeh  4 , Mohamadreza Baghaban Eslaminejad  5 , Hossein Baharvand. Cannabidiol-loaded microspheres incorporated into osteoconductive scaffold enhance mesenchymal stem cell recruitment and regeneration of critical-sized bone defects. Mater Sci Eng C Mater Biol Appl. 2019 Aug;101:64-75.  doi: 10.1016/j.msec.2019.03.070.
  2. Ellen Schmuhl 1 , Robert Ramer 2 , Achim Salamon  3 , Kirsten Peters  3 , Burkhard Hinz. Increase of mesenchymal stem cell migration by cannabidiol via activation of p42/44 MAPK. Biochem Pharmacol. 2014 Feb 1;87(3):489-501.  doi: 10.1016/j.bcp.2013.11.016.
  3. D C Hammell 1 , L P Zhang 2 , F Ma  2 , S M Abshire  2 , S L McIlwrath  2 , A L Stinchcomb  1 , K N Westlund. Transdermal cannabidiol reduces inflammation and pain-related behaviours in a rat model of arthritis. Eur J Pain. 2016 Jul;20(6):936-48.  doi: 10.1002/ejp.818.
  4. Alberto Sainz-Cort 1 , Claudia Müller-Sánchez 2 , Enric Espel. Anti-proliferative and cytotoxic effect of cannabidiol on human cancer cell lines in presence of serum. BMC Res Notes. 2020 Aug 20;13(1):389.  doi: 10.1186/s13104-020-05229-5.
  5. Sinemyiz Atalay 1 , Iwona Jarocka-Karpowicz 1 , Elzbieta Skrzydlewska.  Antioxidative and Anti-Inflammatory Properties of Cannabidiol. Antioxidants (Basel). 2019 Dec 25;9(1):21.  doi: 10.3390/antiox9010021.

We added in the discussion section some phrases at your suggestion. Thank you for your comment!
